# A High-Precision Baseline Calibration Method Based on Estimation of Azimuth Fringe Frequency with THz Interferometry SAR

**Zeyu Wang** [1], **Chao Li** [2,*], **Guohua Zhang** [1], **Shen Zheng** [2], **Xiaojun Liu** [2] and **Guangyou Fang** [2]

[1] Aerospace Information Research Institute, Chinese Academy of Sciences, Beijing 100094, China; wangzeyu20@mails.ucas.ac.cn (Z.W.); zhangguohua21@mails.ucas.ac.cn (G.Z.)

[2] The Key Laboratory of Electromagnetic Radiation and Sensing Technology, Chinese Academy of Sciences, Beijing 100190, China; zhengshen@aircas.ac.cn (S.Z.); lxjdr@mail.ie.ac.cn (X.L.); gyfang@mail.ie.ac.cn (G.F.)

\* Correspondence: cli@mail.ie.ac.cn

**Abstract:** In this study, repeat-pass synthetic aperture radar interferometry (repeat-pass THz InSAR) is first extended to the terahertz band, and it has tremendous potential in the application of high-resolution three-dimensional (3D) imaging due to its shorter wavelength, larger bandwidth, and greater sensitivity to elevation variation. The super-resolution and high sensitivity of THz InSAR pose greater demands on the baseline calibration for high-precision digital elevation model (DEM) generation. To meet the elevation accuracy requirement of THz InSAR, we propose a baseline calibration method relying on the estimation of the azimuth fringe frequency (EAFF) of the interferometric phase. Initially, a model for non-parallel sampling path errors within the squint SAR repeat-pass interferometry was established, and then, we conducted the theoretical analysis of the phase errors induced by the non-parallel errors. Following this, using a reference DEM, the relationship between the fringe frequency of the error phase and the bias in the repeat-path positioning was established. This allowed the estimation of the position errors to be transformed into the frequency spectrum estimation based on the FFT, which would mitigate the impact of unknown SAR sampling positions. Ultimately, we investigated the accuracy of the proposed EAFF calibration method, and the simulation showed that it can achieve the theoretical accuracy when the correlation coefficient exceeds 0.3. Furthermore, we configured the repeat-pass THz InSAR system with the 0.3 THz stepped-frequency radar. Compared to the conventional calibration based on ground control points (GCPs), the 3D reconstruction of both a knife and a terrain model, calibrated using the proposed EAFF algorithm, demonstrated that the elevation accuracy can achieve millimeter-level precision across the entire image swath. The above results also proved the great potential of THz InSAR in high-precision 3D imaging and remote sensing.

**Keywords:** terahertz SAR; repeat-pass interferometry; baseline calibration; fringe frequency estimation; three-dimensional (3D) imaging; remote sensing

## 1. Introduction

The terahertz range lies between the microwave and infrared domain in the spectrum, which generally refers to the band ranging from 0.1 to 10.0 THz. After being extended to SAR, terahertz SAR (THz-SAR) and MIMO-SAR have had a substantial impact on the high-precision imaging and 3D reconstruction fields [1–8]. SAR interferometry (InSAR) is a potent technology for 3D reconstruction in remote sensing, but there is virtually no research on InSAR in the terahertz band as far as we know. In fact, terahertz InSAR holds the potential for high-precision 3D reconstruction owing to its shorter wavelength, broader bandwidth, and greater sensitivity to elevation variation in comparison to the microwave spectrum. Although constrained by the source power at present, terahertz radar is not widely applicable in remote sensing. Through the utilization of a scaled-down

imaging geometry, investigations and explorations into THz InSAR technology can be conducted at close range. This holds substantial implications for the prospective realization of high-precision remote sensing.

Obtaining a high-precision digital elevation model (DEM) is paramount for the study of terahertz InSAR technology. The accuracy of the DEM is influenced by many errors, such as interferometric baseline errors and phase errors [9]. In this context, baseline errors can have a substantial impact on DEM generation, and the challenges associated with the baseline calibration in THz InSAR become more pronounced. Compared to the microwave spectrum, the shorter wavelength of the terahertz wave spectrum leads to a heightened sensitivity to elevation and the increased accuracy of elevation measurements, and this requires stricter baseline calibration accuracy in THz InSAR. The baseline standards impose a stringent precision of the high-frequency pulse sampling positioning of terahertz SAR [10] of at least the sub-millimeter level, which can entail significant expenses when implemented via hardware solutions. Moreover, the baseline always changes along the sampling path, so a single and coarse estimation is insufficient to compensate for the entire elevation error. Last but not least, it is customary for terahertz radar to exhibit a lower signal-to-noise ratio compared to the microwave band [11,12].

Although research on THz InSAR is limited, substantial research has been devoted to the advancement of baseline calibration algorithms in microwave InSAR. The calibration methods can be classified into two primary approaches. The first approach employs ground control points (GCPs) in iterative procedures, with a primary focus on utilizing the sensitivity matrix [13–15]. The accuracy of baseline calibration based on GCPs is intricately linked to the level of phase noise [16], and given the more-severe noise corruption in the THz band, which can result in millimeter-level discrepancies, it is deemed inadequate for high-accuracy DEM generation. The second method establishes the relationship between the interferometric phase and baseline configuration, facilitating the calculation of baseline errors using observed phase information [17–19]. For similar outcomes, the estimation accuracy based on the local window and wavenumber shift in the range direction is limited, and thus, it cannot address the minor changes in the baseline that occur as the radar operates. The external DEM can also be used in the baseline calibration [16,20]. It is important to gain an accurate DEM to obtain high-precision baseline calibration.

To solve the problem of the high-accuracy baseline estimation mentioned above, the baseline calibration method relying on the estimation of the azimuth fringe frequency (EAFF) of the interferometric phase is proposed. A model for non-parallel sampling path errors within the squint SAR repeat-pass interferometry was established, and then, the theoretical analysis of the phase errors induced by the non-parallel errors was conducted. Following this, using a reference DEM, the relationship between the azimuth fringe frequency of the error phase and the bias in the repeat-path positioning was established. This allowed the estimation of the baseline errors to be transformed into the frequency spectrum estimation based on the Fast Fourier Transform (FFT), which sidesteps the unknown SAR sampling positions. Then, taking into account the theoretical elevation resolution within the terahertz band, the parameter accuracy with the FFT is provided in detail. Consequently, the proposed EAFF calibration algorithm enabled global interference phase compensation and guaranteed the generation of a higher-precision DEM.

Compared to the enhancement of the precision positioning system for THz SAR, trajectory errors were compensated by the EAFF through software-based solutions, offering the advantages of efficiency and cost-effectiveness. Compared to other calibration methods, the proposed EAFF algorithm effectively integrates the correlation between the sampling trajectory deviations and phase errors. These enhancements effectively convert the challenge related to minute baseline changes into a precise frequency estimation. Utilizing the exceptional sensitivity to elevation of THz InSAR, it becomes easier to achieve spectrum analysis in the frequency domain. Meanwhile, exploiting its high resolution, the EAFF method enables a significantly elevated level of precision in the baseline estimation for THz InSAR. According to the detailed formulations, the adjustment of the FFT sampling

rates ensures high-precision trajectory error and baseline estimation, meeting the demand for superior elevation resolution. An ancillary advantage of this method is its operational efficiency, as it eliminates the need for iterative algorithms.

This article is organized as follows. Section 2 performs the principle study of InSAR, including the high-precision model, the corresponding height error analysis, and the coherence analysis of the terahertz band, which determines the elevation accuracy of THz InSAR. Section 3 investigates the principles, implementation process, and parameter accuracy of the EAFF methods in detail. In Section 4, we perform the simulation to estimate the practical accuracy under various noise levels, which serves as a reference for the experiments. Section 5 presents the experiment using the repeat-pass THz InSAR with a 0.3 THz stepped-frequency radar. The results of both the knife and terrain models demonstrate that the elevation accuracy can achieve the millimeter level within the whole swath coverage. Section 6 concludes the entire paper and outlines the directions for further research.

## 2. Theory of High-Precision THz InSAR Model

### 2.1. The Principle of SAR Interferometry

Figure 1 shows the classic InSAR geometry with baseline $B$, inclined at $\alpha$. It is well known that the information of the phase is of great interest for InSAR, and the interferometric phase $\phi$ of the "ping-pong" mode is

$$\phi = -\frac{4\pi}{\lambda}(R_1 - R_2) = \frac{4\pi}{\lambda}\Delta R \tag{1}$$

where $\lambda$ is the radar wavelength, $R_1$ and $R_2$ refer to the distances from one target to antennas $A_1$ and $A_2$, respectively, and $P$ is an imaginary point whose altitude is $h$ with respect to the ground, and its corresponding reference point is $P_0$. The look angle $\theta$ is related to the point height $h$, and the difference of the look angle of $P$ and $P_0$ is $\delta\theta$. According to the geometric relationship between the slope distance difference $\Delta R$ and $B$, one can obtain

$$
\begin{aligned}
\Delta R &= -B\sin(\theta + \delta\theta - \alpha) \\
&= -B\sin(\theta - \alpha)\cos(\delta\theta) - B\cos(\theta - \alpha)\sin(\delta\theta) \\
&\approx -B\sin(\theta - \alpha)(1 - \frac{\delta\theta^2}{2}) - B\cos(\theta - \alpha)(\delta\theta - \frac{\delta\theta^3}{6})
\end{aligned}
\tag{2}
$$

With the help of Taylor expansion, the trigonometric functions in (2) can be approximated as the combination of the polynomials to simplify the equation. Unlike the conventional interferometric model, a higher-order approximation is used in the high-precision 3D close-range imaging scenario. Because the radar look angle varies greatly in the near-field, the first-order approximation generates a large approximation error, which causes a huge height error and reduces the height resolution.

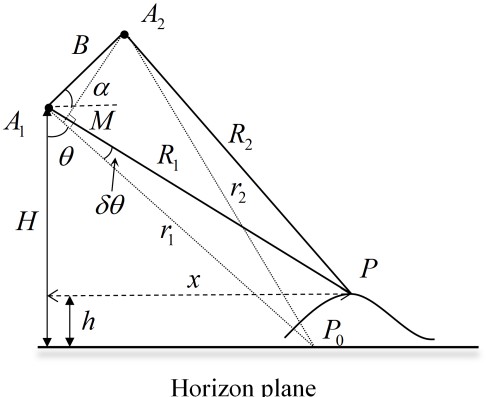

**Figure 1.** Cross-track SAR interferometry imaging geometry.

Generally speaking, $\phi$ consists of the terrain phase and the reference phase. Using the concept of the wavenumber, which is $4\pi/\lambda$ in "ping-pong" mode, (2) can be transformed into the formation of the phase difference, and its important is in removing the reference phase because it does not contribute to the extraction of the altitude, but aggravates the wrapped phase. Thus, the terrain phase is as follows:

$$\frac{\phi_h}{kB} = -\cos(\theta - \alpha)\delta\theta - \sin(\theta - \alpha)\frac{\delta\theta^2}{2} + \cos(\theta - \alpha)\frac{\delta\theta^3}{6} \tag{3}$$

By transforming (3) into a standard cubic equation, the relationship between $\delta\theta$ and the terrain phase $\phi_h$ can be solved, expressed as $\delta\theta(\phi_h)$. The detailed procedure is described in the Appendix A. To date, we have identified the approach to transforming the interferometric phase into $\delta\theta$; however, it is still necessary to establish the connection between $\delta\theta$ and the target altitude $h$. Based on the geometry of the antennas and targets shown in Figure 1, one can obtain

$$\begin{cases} \cos\theta = H/R_1 \\ \cos(\theta + \delta\theta) = (H - h)/R_1 \end{cases} \tag{4}$$

and

$$\begin{aligned} \cos(\theta + \delta\theta) &\approx \cos\theta(1 - \frac{\delta\theta^2}{2}) - \sin\theta(\delta\theta - \frac{\delta\theta^3}{6}) \\ &= \cos\theta(1 - \frac{\delta\theta^2}{2}) - \sqrt{(1 - \cos^2\theta)}(\delta\theta - \frac{\delta\theta^3}{6}) \end{aligned} \tag{5}$$

By substituting (4) to (5), one can obtain

$$h(\delta\theta) = -\frac{\sqrt{R_1^2 - H^2}}{6}\delta\theta^3 + \frac{H}{2}\delta\theta^2 + \sqrt{R_1^2 - H^2}\delta\theta \tag{6}$$

By replacing can be estimated from the interferogram. The height sensitivity is usually defined as the height variation when $\delta\theta = 2\pi$. Due to the shorter wavelength, the height sensitivity of THz InSAR is much smaller than that of microwave InSAR.

To demonstrate the efficacy of our model, the simulations of the elevation estimation using both the first-order and higher-order approximations are conducted. The main antenna is positioned at an elevation of 0.32 m, and the target is situated at an distance of 3 m. As shown in Figure 2, both two approximation models provide good fits to the theoretical values, especially when the target is relatively low. However, as the target's altitude increases, the first-order model exhibits noticeable estimation errors, leading to a significant reduction in the height accuracy. Conversely, the higher-order model yields accurate predictions. Proved by the simulations, the relative height errors are minimized using the higher-order approximation model, demonstrating its exceptional accuracy and resilience.

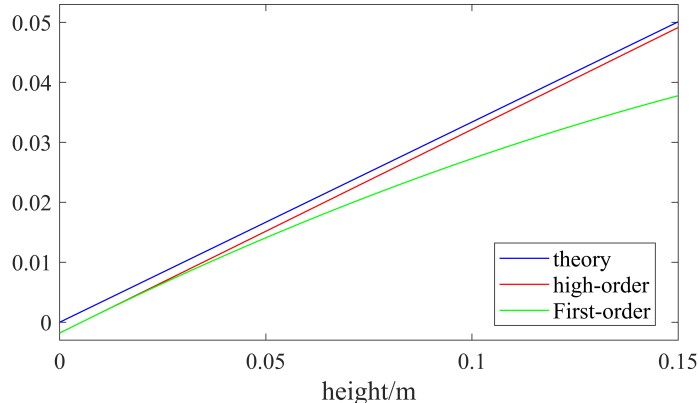

**Figure 2.** Results of the height estimation of different approximations.

### 2.2. The Analysis of Height Errors Based on InSAR Parameter Errors

After refining the interferometric model in the preceding subsection, the height errors resulting from various factors are analyzed in the following paragraphs. It is essential to determine an achievable elevation accuracy. As shown in Figure 1, for point $P$, the expression of its height $z$ and distance $y$ can be obtained by

$$
\begin{aligned}
z &= H - R_1 \cos\theta = H - R_1 \cos(\theta - \alpha + \alpha) \\
&= H - R_1 \left[ \sqrt{1 - \sin^2(\theta - \alpha)} \cos(\alpha) - \sin(\theta - \alpha) \sin(\alpha) \right]
\end{aligned}
\tag{7}
$$

$$
y = R_1 \sin\theta
\tag{8}
$$

By applying the Law of Cosines in the triangle $\Delta PA_1 A_2$, the relationship among $B$, $R_1$, $\theta$, and $\alpha$ can be derived as follows:

$$
\sin(\theta - \alpha) = \frac{-(2R_1 + \Delta R)\Delta R + B^2}{2R_1 B}
\tag{9}
$$

where the offset $\Delta R = \phi / k$, and $\phi$ represents the phase difference . By the way, both $\phi$ and $\Delta R$ are less than 0 because of their definitions.

The sources of elevation errors can be classified into directness and indirectness, as described in Reference [21]. The instinct error refers to the error directly caused by the model itself, while the indirect error refers to the one caused by location errors indirectly affecting the estimation. The instinct errors include the estimation errors of a baseline $B$, a tilt angle $\alpha$, an interferometric phase $\phi$, an antenna height $H$ , and a slant range $R_1$. We express them by $\delta z_B$, $\delta z_\alpha$, $\delta z_\phi$, $\delta z_H$, and $\delta z_{R_1}$, respectively. $\tau_x$ and $\tau_y$ represent the tilt angles in the azimuth and distance directions. If a distance position error is $\delta y$, the corresponding altitude error will be $\delta z_{\delta y} = \tan(\tau_y)\delta y$. By combining (7)–(9), the elevation errors related to the aforementioned factors are expressed by

$$
\delta z_B = (\sin(\theta) + \tan\tau_x \cos(\theta)) \cdot \left( -\frac{R_1 \tan(\theta_1 - \alpha)}{B} + \frac{1}{\cos(\theta_1 - \alpha)} \right) \delta B
\tag{10}
$$

$$
\delta z_\alpha = R_1(\sin\theta + \tan\tau_x \cos\theta)\delta\alpha
\tag{11}
$$

$$
\delta z_\phi = -\frac{R_1(\sin(\theta_1) + \tan\tau_x \cos(\theta))}{kB\cos(\theta - \alpha)} \frac{R_1 + \Delta r}{R_1} \delta\varphi
\tag{12}
$$

$$
\delta z_H = \delta H
\tag{13}
$$

$$
\delta z_{R_1} = -\cos\theta\delta R_1
\tag{14}
$$

Quantifying each error will be advantageous for the calculation of the elevation accuracy. On the ond hand, the quantified errors associated with most parameters can be regarded as the systemic biases which typically exhibit nearly constant deviations within the imaging range. Among these biases, the baseline bias, including $\delta B$ and $\delta\alpha$, significantly impacts the height accuracy. On the other hand, the phase error plays a crucial role in determining the height accuracy, as it is closely connected to a relative height variation. Minimizing the interferometric phase error, as outlined in (12), can yield an optimal imaging scheme. Subsequently, the following subsection will furnish a comprehensive analysis of the coherence of InSAR imaging.

### 2.3. The Coherence of InSAR System

The expression of the impulse response function of SAR is [22]:

$$
W(x,y) = sinc\left(\frac{\pi x}{R_x}\right) sinc\left(\frac{\pi y}{R_y}\right)
\tag{15}
$$

where $R_x$ and $R_y$ are the azimuth and range resolutions, respectively. In InSAR, the return waveforms $s_1$ and $s_2$ of a point $(x_0, y_0)$ are

$$s_1(x_0, y_0) = \iint f(x, y) \exp\left(-j\frac{4\pi R_1}{\lambda}\right) \cdot W(x - x_0, y - y_0)dxdy + n_1 \tag{16}$$

$$s_2(x_0, y_0) = \iint f(x, y) \exp\left(-j\frac{4\pi R_2}{\lambda}\right) \cdot W(x - x_0, y - y_0 + \delta y)dxdy + n_2 \tag{17}$$

where $f(x, y)$ is a two-dimensional complex scattering function of scatters. $R_1$ and $R_2$ represent the slant ranges from the target to the two antennas, and an image registration error $\delta y$ is introduced. $n_1$ and $n_2$ represent the Gaussian white-noise in two channels. It is assumed that the complex scattering functions are the zero-mean two-dimensional complex Gaussian white noise with the same root mean square, and their cross-correlation function is [21]

$$\langle f(x, y)f^*(x', y')\rangle = \delta(x - x', y - y')\sigma_0 \tag{18}$$

Therefore, the complex cross-correlation function of the main and auxiliary images can be expressed as

$$\langle s_1 s_2^*\rangle = \iiiint f(x_1, y_1)f^*(x_2, y_2) \exp\left(-j\frac{4\pi(R_1 - R_2)}{\lambda}\right)$$
$$\cdot W(x_1 - x_0, y_1 - y_0)W(x_2 - x_0, y_2 - y_0 + \delta y)dx_1 dy_1 dx_2 dy_2 \tag{19}$$

By substituting (18) into (19), the cross-correlation coefficient of the same pixel can be obtained as

$$\langle s_1 s_2^*\rangle = \sigma_0 \iint \exp\left(-j\frac{4\pi(R_1 - R_2)}{\lambda}\right)W(x - x_0, y - y_0)W(x - x_0, y - y_0 + \delta y)dxdy \tag{20}$$

The slant range difference is approximated as [21]

$$R_1 - R_2 \approx B\sin(\theta_0 - \alpha) - \frac{B_\perp}{r_0}(y - y_0)\cos\theta_0 \tag{21}$$

$$B_\perp = B\cos(\theta_0 - \alpha) \tag{22}$$

By replacing (21) and (22) into (20), one can obtain

$$\langle s_1 s_2^*\rangle = \sigma_0 R_x \exp\left(-j\frac{4\pi}{\lambda}B\sin(\theta_0 - \alpha)\right)\int \exp\left(j\frac{4\pi}{\lambda}\frac{B_\perp}{r_0}(y - y_0)\cos\theta_0\right)$$
$$\cdot \frac{\sin\pi(y - y_0)/R_y}{\pi(y - y_0)/R_y}\frac{\sin\pi(y - y_0 + \delta y)/R_y}{\pi(y - y_0 + \delta y)/R_y}dy \tag{23}$$

Integrating (23) with respect to $y$, one can obtain

$$\langle s_1 s_2^*\rangle = \sigma_0 R_x R_y \exp\left(-j\frac{4\pi}{\lambda}B\sin(\theta_0 - \alpha)\right)\alpha \tag{24}$$

$$\alpha = \left(1 - |\alpha_y|R_y\right)\exp(j\pi\alpha_y\delta y)\frac{\sin\pi\delta y(1 - |\alpha_y|R_y)/R_y}{\pi\delta y(1 - |\alpha_y|R_y)/R_y}, \quad \alpha_y = \frac{kB_\perp}{2\pi r_0\tan\theta_0} \tag{25}$$

Considering that the auto-correlation functions of both images are as follows [23]

$$\langle s_1 s_1^*\rangle = \sigma_0 R_x R_y + N, \quad \langle s_2 s_2^*\rangle = \sigma_0 R_x R_y + N \tag{26}$$

where $N = \langle n_1^2 \rangle = \langle n_2^2 \rangle$. Then, the correlation coefficient of InSAR is expressed as

$$\gamma = \frac{|\langle s_1 s_2^* \rangle|}{\sqrt{\langle s_1 s_1^* \rangle \langle s_2 s_2^* \rangle}} = \frac{|\alpha|}{1 + SNR^{-1}} \tag{27}$$

where SNR is the signal-to-noise rate. Assuming that $\delta y = 0$, the correlation coefficient $\gamma$ can be expressed as

$$\gamma = \gamma_{SNR} \cdot \gamma_{surface} \tag{28}$$

$$\gamma_{SNR} = \frac{1}{1 + SNR^{-1}} \tag{29}$$

$$\gamma_{surface} = 1 - \frac{kB_\perp R_y}{2\pi R_0 \tan\theta_0} \tag{30}$$

Considering a single-look situation, the standard deviation of the interferometric phase related to $\gamma$ obeys [24]

$$\sigma_{\Delta\varphi} = \sqrt{\frac{1 - \gamma^2}{2\gamma^2}} \tag{31}$$

It can be inferred that a smaller $\gamma$ leads to more severe phase noise in the interferogram, according to (31). However, there have been studies focusing on the phase noise caused by the geometric decorrelation. The reference [17] pointed out that the relative shifts of the ground wavenumber spectra occurred in InSAR images. Based on that, a pre-filtering method was proposed for an improvement of the interferogram quality. The phase noise caused by the geometric decorrelation could be filtered significantly. Moreover, the reference [25] focused on the phase noise of multilook InSAR resulted from the geometric decorrelation. A spectral model relying on the spectra of the individual channels and multilook averaging windows in range was proposed. The results showed that this model aligned well with the numerical simulations.

## 3. Processing of the EAFF Calibration Method

### 3.1. The Establishment of Models and Theoretical Framework

For InSAR imaging, the elevation is obtained based on the phase difference caused by the difference in slant ranges resulting from different imaging geometries. For both single-pass and repeat-pass interferometry, the position difference of two antennas constitutes the baseline. The variable baseline errors at different azimuth positions will result in height errors in cross-track interferometry. If the calibration precision of the baseline is deficient, it can introduce substantial elevation inaccuracies, consequently resulting in a marked reduction of 3D reconstruction.

Conventional InSAR depends on the inertial navigation systems or the Global Positioning System (GPS) to acquire the positions of sampling points [26,27]. In our particular sampling context, the vehicle's wheel controls the emitting and receiving pulses at pre-determined intervals. However, given the higher baseline accuracy of THz InSAR, it necessitates a heightened level of positioning precision on the order of millimeters or even sub-millimeters. Implementing hardware systems for such precision would significantly increase the cost. Consequently, we mitigate baseline errors that arise during the two scanning procedures through the innovative EAFF calibration algorithm.

The theoretical data acquisition trajectories for a specific baseline should remain parallel during a sampling process. In practice, the imaging trajectory invariably exhibits rotations and offsets. Let $dx$, $dy$, and $dz$ represent deviations in the azimuth, distance, and altitude, respectively. Also, $\theta_z$, $\theta_x$, and $\theta_y$ denote rotation angles around the axis of height, azimuth and distance, in that order. Figure 3 depicts the schematic representation of the target localization, trajectory rotations, and translation offsets projected onto the x–y plane. The two solid black lines with arrows signify the actual sampling paths, while the black dashed line represents the theoretical one paralleled to the other. In the coordinate

system of the main trajectory, the points $T_1$ and $T_2$ correspond to the same distances but different azimuths. The angle $\theta_z$ leads to noticeable changes in the slant ranges of these points, resulting in distinct phase differences between them. Furthermore, as their azimuths diverge, the slant range errors become more pronounced. Similarly, for points $T_1$ and $T_3$ sharing the same azimuths, the slant range errors also manifest due to differences in distances.

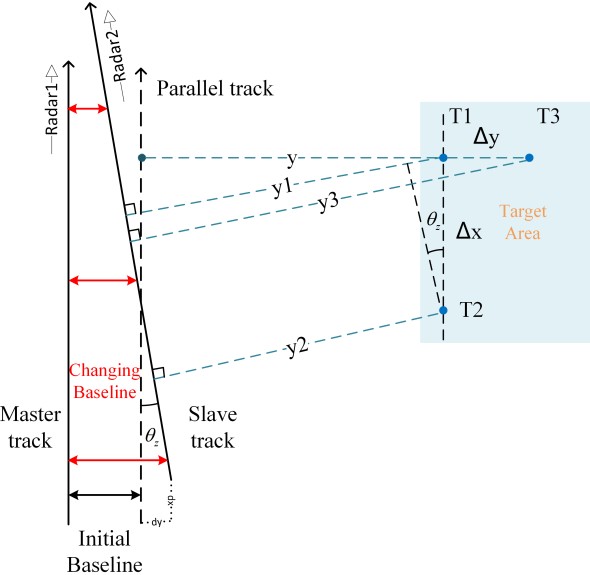

**Figure 3.** The geometric diagram of x–y plane projection of repeat-pass track after shifted by $\theta_z$.

Next, it is time to analyze in detail the impact of the various track error parameters. Taking the theoretical radar sampling path as the imaging coordinate system's x-axis, for a point $\vec{P} = (x, y, z)^{\mathrm{T}}$, its theoretical SAR coordinates are

$$\vec{P} = (X, R)^{\mathrm{T}} = \left( x, \sqrt{y^2 + z^2} \right)^{\mathrm{T}} \tag{32}$$

On the one hand, when the translation errors occur, the coordinates can be represented as

$$\vec{P}'_{trans} = \begin{bmatrix} x'_{trans} \\ y'_{trans} \\ z'_{trans} \end{bmatrix} = \begin{bmatrix} x \\ y \\ z \end{bmatrix} + \begin{bmatrix} dx \\ dy \\ dz \end{bmatrix} = \begin{bmatrix} x + dx \\ y + dy \\ z + dz \end{bmatrix} \tag{33}$$

and the corresponding SAR coordinates are

$$\vec{P}_{trans} = \begin{bmatrix} x'_{trans} \\ \sqrt{(y'_{trans})^2 + (z'_{trans})^2} \end{bmatrix} = \begin{bmatrix} x + dx \\ \sqrt{(y + dy)^2 + (z + dz)^2} \end{bmatrix} \tag{34}$$

According to (34), the first-order sensitivity matrix for three-dimensional coordinates is as follows

$$F_T = \begin{bmatrix} \frac{\partial \vec{P}}{\partial x} & \frac{\partial \vec{P}}{\partial y} & \frac{\partial \vec{P}}{\partial z} \end{bmatrix} = \begin{bmatrix} \frac{\partial X}{\partial x} & \frac{\partial X}{\partial y} & \frac{\partial X}{\partial z} \\ \frac{\partial R}{\partial x} & \frac{\partial R}{\partial y} & \frac{\partial R}{\partial z} \end{bmatrix} = \begin{bmatrix} 1 & 0 & 0 \\ 0 & \frac{y}{R} & \frac{z}{R} \end{bmatrix} \tag{35}$$

Therefore, the positioning errors caused by the translation offset $\Delta \vec{P}_{trans}$ are

$$\Delta \vec{P}_{trans} = \begin{bmatrix} \Delta X_{trans} \\ \Delta R_{trans} \end{bmatrix} = F_T \cdot \begin{bmatrix} dx \\ dy \\ dz \end{bmatrix} = \begin{bmatrix} dx \\ \frac{y}{R} dy + \frac{z}{R} dz \end{bmatrix} \tag{36}$$

On the other hand, the coordinates after undergoing a triple angle rotation can be expressed as

$$\vec{P}'_{rot} = R \cdot \vec{P} \tag{37}$$

where the rotation matrix $R$ can be represented as

$$R = R_Z(\theta_z) R_X(\theta_x) R_Y(\theta_y) \tag{38}$$

$$R_Z(\theta_z) = \begin{bmatrix} \cos\theta_z & -\sin\theta_z & 0 \\ \sin\theta_z & \cos\theta_z & 0 \\ 0 & 0 & 1 \end{bmatrix}, R_X(\theta_x) = \begin{bmatrix} 1 & 0 & 0 \\ 0 & \cos\theta_x & -\sin\theta_x \\ 0 & \sin\theta_x & \cos\theta_x \end{bmatrix}, R_Y(\theta_y) = \begin{bmatrix} \cos\theta_y & 0 & \sin\theta_y \\ 0 & 1 & 0 \\ -\sin\theta_y & 0 & \cos\theta_y \end{bmatrix} \tag{39}$$

and the SAR coordinates are

$$\vec{P}_{rot} = \left( x'_{rot}, \sqrt{(y'_{rot})^2 + (z'_{rot})^2} \right) \tag{40}$$

After rotations, the coordinates become more intricate. By taking into account the experimental conditions, we can eliminate specific interfering terms to improve the practicality and pertinence of the rotation model. Given that the x-axis aligns with the radar's motion direction, and $\theta_x$ corresponds to changes in the radar's look angle. According to the radar positioning theory, for a target, both its slant range and azimuth remain constant when the look angle varies. Therefore, we assume that $R_X = I$, and $I$ is the unit matrix. $\theta_y$ characterizes the variations in terrain between two scans. In the experiment, we adopt a method of constructing the baseline through a scene translation, and further details will be provided in the experimental section. The radar's position exhibits minimal deviations. We assume that $R_Y = I$, given the flat terrain and relatively short imaging distances. Based on the analysis above, in the experiment, $\theta_z$ makes a highest impact on the height accuracy. Therefore, we focus on addressing the estimation problem related to $\theta_z$. Due to $\theta_z$, the position variations are

$$\Delta\vec{P}_{\theta_z} = \begin{bmatrix} \Delta X_{\theta_z} \\ \Delta R_{\theta_z} \end{bmatrix} = \begin{bmatrix} x(\cos\theta_z - 1) - y\sin\theta_z \\ \frac{y}{R}[y(\cos\theta_z - 1) + x\sin\theta_z] \end{bmatrix} \tag{41}$$

As (41) shows, the slant range difference $\Delta R_{\theta_z}$ is related to three coordinates and the trajectory offset angle $\theta_z$. The larger value of these variables, the more severe deviation will be. This leads to the undesirable error difference phase (EDP)

$$EDP = k \cdot \Delta R_{\theta_z} \tag{42}$$

in the azimuth direction obviously, due to the large wavenumber $k$. For example, if $\theta_z = 0.5°$, an EDP with different imaging centers is shown in Figure 4. The values are rather close when the target's distances are 3 m, 5 m, and 7 m. However, the phase wrapping occurs according to the rising azimuth internal between two targets. For the discrete corner reflectors, it becomes more difficult to calculate an integer period of wraps when $\theta_z$ is relatively large. Unfortunately, even with only a 0.5° deviation, there is still significant EDP in small imaging ranges.

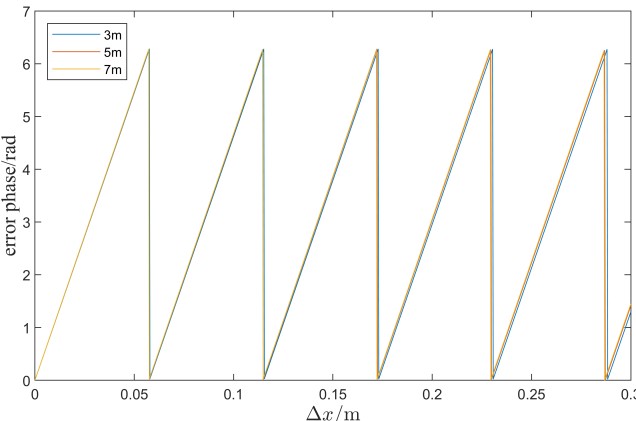

**Figure 4.** EDF corresponding to varying azimuth interval $\Delta x$ in three range centers.

*3.2. The Processing and Resolution Analysis of the EAFF Method*

The preceding section delved into the origins of errors, resembling the error compensation. In this subsection, we delineate the error estimation procedure and the precision of each parameter. Given the high sensitivity of EDP to $\Delta x$, we establish a connection between the rate of phase variation in a reference DEM and $\theta_z$. Thus, $\theta_z$ can be computed by estimating the fringe frequency in the interferogram. When a minor azimuth displacement $\Delta x$ arises, one can derive

$$\frac{4\pi}{\lambda}\Delta R_{\theta_z} = 2\pi f_x \Delta x \tag{43}$$

where $f_x$ is the fringe frequency of the interference phase in the azimuth direction. Actually, if $\theta_z = 1°$, there will be $\sin(1°) = 0.0175$ and $\cos(1°) - 1 = -1.5230 \times 10^{-4}$. Due to the fact that $(\cos\theta_z - 1) \ll \sin\theta_z$, we can ignore the cosine impact. By combining (41)–(43), one can obtain

$$\sin\theta_z = \frac{\lambda}{2} f_x \frac{R}{y} \tag{44}$$

Therefore, by estimating the azimuth fringe frequency of the calibration plane, the offset angle of the trajectory can be obtained. Then, based on the relationship among the slant range error and the various parameters as described in (42) and (43), the interference phase can be calculated. Compensating the interference phase back into the interferogram allows for the higher height accuracy. It is worth noting that the sampling rate must be at least twice the frequency of the signal according to Nyquist's theorem. Following (44), $\theta_z$ and $f_x$ are directly proportional. It is necessary for $|f_x| < 1/2$ to be satisfied, implying that $\theta_z < \arcsin(\lambda R / 4y)$ must also be fulfilled. This indicates that the EAFF algorithm is more suitable for estimating precise deviations.

According to (41), it is evident that a broader imaging range results in larger phase errors. For a determined height accuracy, this implies greater demand for the precision of $\theta_z$ for larger image scales. As a result, we establish a correlation between the image ranges and parameter accuracies. Assuming the range in azimuth direction is denoted as $L_x$, the maximum EDF within this range can be expressed as follows:

$$\text{EDP} = \frac{ky}{R} L_x \sin\theta_z \tag{45}$$

Thus, we can obtain that within the given imaging range, if the acceptable phase error is $\rho_{\Delta\varphi}$, the resolution of $\theta_z$ will be

$$\rho_{\theta_z} = \arcsin\left(\frac{\rho_{\Delta\varphi}R}{kL_x y}\right) \tag{46}$$

Then, the baseline accuracy caused by $\rho_{\theta_z}$ is

$$\rho_B = \frac{\rho_a \sin(\rho_{\theta_z})}{\cos \alpha} \tag{47}$$

where $\rho_a$ represents the radar's azimuthal resolution. According to (44), we can obtain the frequency resolution as

$$\rho_{f_x} = \frac{ky}{2\pi R} \sin(\rho_{\theta_z}) = \frac{\rho_{\Delta\varphi}}{2\pi L_x} \tag{48}$$

Here, the 2D FFT transformation is applied to estimate the fringe frequency. By implementing the 2D-FFT on the interference phase of the calibration plane, represented by $s_p(\tau, \eta)$, the fundamental frequency in the azimuth direction, i.e., the fringe frequency of the interference phase, can be obtained

$$\hat{f}_x = \arg \max_{f_x}(abs(FFT(s_p(\tau, \eta)))) \tag{49}$$

where $abs()$ is the operation of taking absolute values. This enables the acquisition of the correct interferometric phase. If the sampling frequency $F_s$ is known, then the number of sampling points $N_s$ should satisfy the following

$$N_s > \frac{F_s}{\rho_{f_x}} \tag{50}$$

In summary, we can delineate the complete algorithmic processing steps, as illustrated in the flowchart presented in Figure 5. Furthermore, we can employ both the forward and reverse cognitive approaches to assess the elevation accuracy. The forward thinking centers on attaining the necessary parameter accuracy for the algorithmic processing, guided by the elevation accuracy of an entire scene. This approach proves valuable in an image scheme design, allowing the establishment of calibration precision aligned with the desired height accuracy. The reverse cognition involves gauging the achievable elevation accuracy based on the estimated accuracy of parameters, and it serves as the foundation for the global height error estimation.

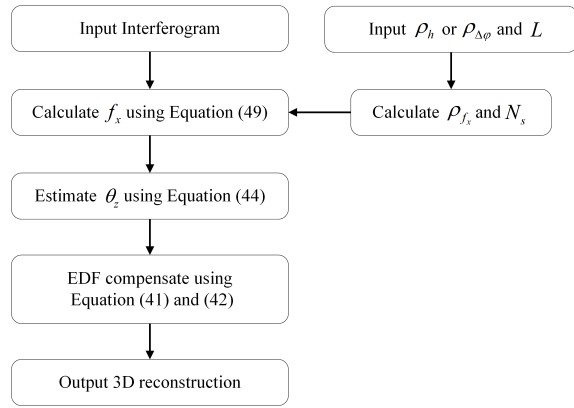

**Figure 5.** Workflow of the EAFF calibration algorithm. Given the desirable accuracy $\rho_h$ or $\rho_{\Delta\varphi}$ and swath range $L$ for DEM generation, the parameter can be exact calculated to guide the EAFF algorithm.

## 4. The Simulation for EAFF Baseline Calibration Accuracy

Section 3 provides a detailed analysis of the principles, implementation process, and estimation accuracy analysis of the calibration method based on the fringe frequency estimation. The noise intensity is related to the coherence coefficient, according to (31). Therefore, we need to simulate different levels of the covered noise to assess whether the estimation accuracy can meet the theoretical requirement. Considering that actual interferometric phase

images may be significantly affected by the white noise, the simulations introduce a flat model and incorporates the interference phase noiseThe phase errors consist of EDF caused by rotation angles and the Gaussian white noise with varying thermal noise intensity. The simulated parameters are as shown in Table 1.

**Table 1.** Simulation parameters of FFT-based EDF estimation.

| Parameter | Symbol | Value | Unit |
|---|---|---|---|
| Range sampling number | $N_{fast}$ | 64 | - |
| Azimuth sampling number | $N_{slow}$ | 64 | - |
| Range resolution | $\rho_r$ | 0.005 | m |
| Azimuth resolution | $\rho_a$ | 0.005 | m |
| Wavelength | $\lambda$ | 0.001 | m |
| Radar look angle | $\theta$ | 75 | ° |
| Radar height | $H$ | 0.33 | m |
| Baseline length | B | 0.1 | m |
| Baseline tilted angle | $\alpha$ | 0 | ° |
| Height ambiguous | $\Delta h$ | 0.0238 | m |
| Rotation angle accuracy | $\rho_{\theta_z}$ | 0.001 | rad |
| Frequency accuracy | $\rho_{f_x}$ | 1.93 | Hz |
| Up-sampling number | $N_s$ | 128 | - |

Figure 6 provides the noisy phase images corresponding to the different coherence coefficients when $\theta_z = 0.0087$ rad . Figure 6a is obtained by calculating (41) and (42) without the white noise. Consistent with the line chart in Figure 4, the conspicuous wrapping fringes are observed in the azimuth direction, whereas they remain nearly unchanged in the range direction. Figure 6b–f depict increasing coherence coefficients, leading to the progressively clearer fringes. The analysis suggests that there is probably a lowest limit for the noise intensity. It is necessary to simulate the relationship between $\rho_{f_x}$ and $\gamma$ to avoid the situation where the algorithm is failed because of the white noise.

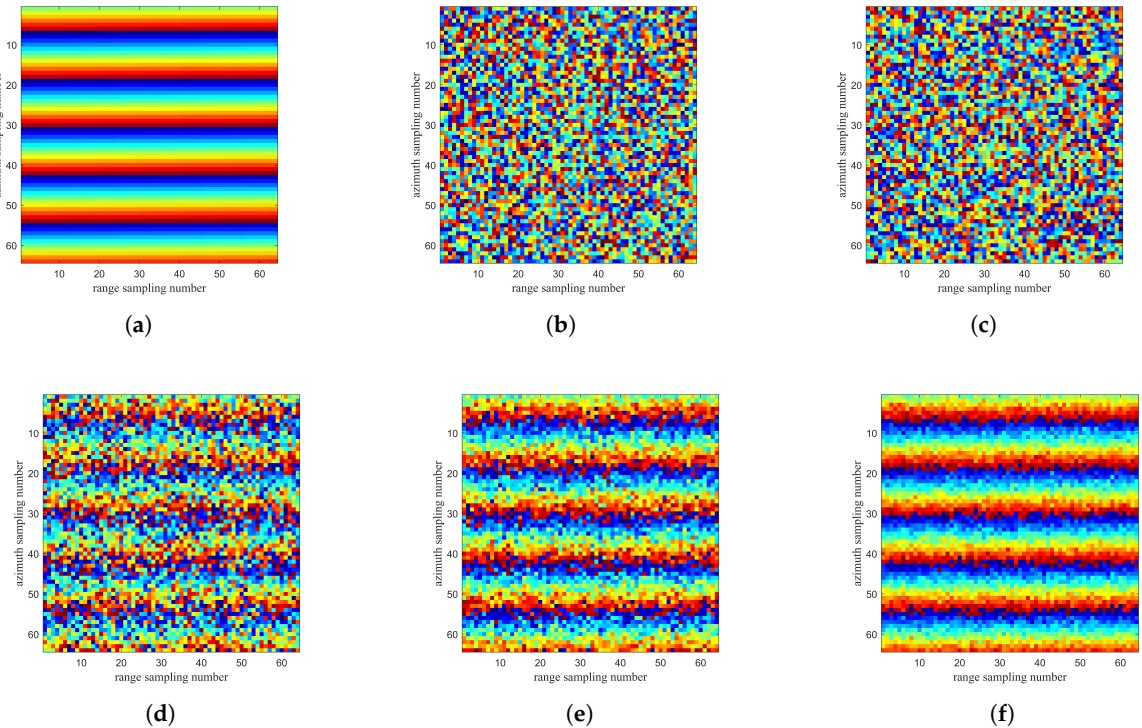

**Figure 6.** The interferometric fringe formed by $\theta_z = 0.0087$ rad with different $\gamma$. (**a**) $\gamma = 1$. (**b**) $\gamma = 0.1$. (**c**) $\gamma = 0.3$. (**d**) $\gamma = 0.5$. (**e**) $\gamma = 0.7$. (**f**) $\gamma = 0.9$.

Figure 7 displays the estimation phase errors for three rotational angles, each with varying coherence coefficients. In general, considering an accuracy of 0.001 rad, the FFT-based estimation satisfies the requirements for all three rotation angles when $\gamma > 0.3$. The fringe frequency appears to exert no substantial influence on the estimation accuracy, as long as the sampling number is compliant. The line chart indicates that the slightly higher fringe frequency offers an advantage for the estimation. Reference [28] also concluded that the error estimation of the fringe was irrelevant to its center frequency, by analyzing the comparison of various estimation methods such as the complex signal phase derivative (CSPD) and the maximum likelihood (ML).

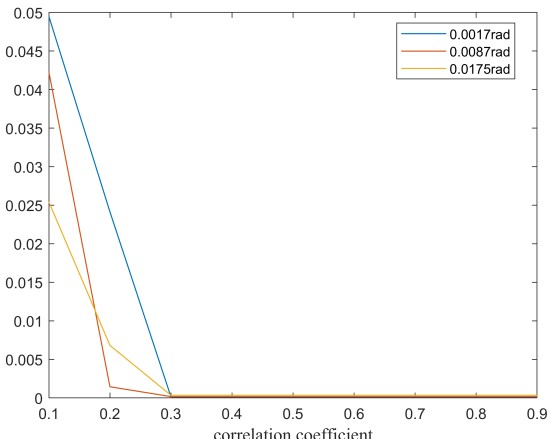

**Figure 7.** The result of phase error estimation under varying levels of white noise for three $\theta_z$.

In addition, the planarity of the surfaces will exert an influence on the rotation angle accuracy. Referring to (41), targets' coordinates have an impact on the slant range. By combining (42) and (43), the relationship between $f_x$ and the coordinates is given by

$$f_x = \frac{2}{\lambda}\left[\frac{y^2}{xR}(\cos\theta_z - 1) + \frac{y}{R}\sin\theta_z\right] \tag{51}$$

The partial derivatives of $f_x$ with respect to the three coordinates are

$$\frac{\partial f_x}{\partial x} = -\frac{2y^2}{x^2\lambda R}(\cos\theta_z - 1) \tag{52}$$

$$\frac{\partial f_x}{\partial y} = \frac{2}{\lambda}\left[\frac{2y}{xR}(\cos\theta_z - 1) + \frac{1}{R}\sin\theta_z\right] \tag{53}$$

$$\frac{\partial f_x}{\partial z} = -\frac{2z}{\lambda R^3}\left[\frac{y^2}{x}(\cos\theta_z - 1) + y\sin\theta_z\right] \tag{54}$$

Assume that the deviations in the azimuth, range and altitude directions are represented by $dx$, $dy$, and $dz$, respectively. The frequency shift can be obtained by

$$\Delta f_x = \frac{\partial f_x}{\partial x}dx + \frac{\partial f_x}{\partial y}dy + \frac{\partial f_x}{\partial z}dz \tag{55}$$

Combining with (44), the estimation error of the rotation angle is given by

$$\Delta\theta_z = \frac{\partial\theta_z}{\partial f_x}\Delta f_x = \frac{\lambda R}{2y|\cos\theta_z|}\Delta f_x \tag{56}$$

Figure 8 shows that the trend of first-order sensitivity coefficients. With the increasing distance, the impacts of distance and height coefficients tend to diminish, while the azimuth coefficient shows an increasing influence. The numerical analysis reveals that the distance

coefficient has the most pronounced effect. Advantageously, owing to the system's ultra-high resolution, the distance errors are confined to the millimeter scale.

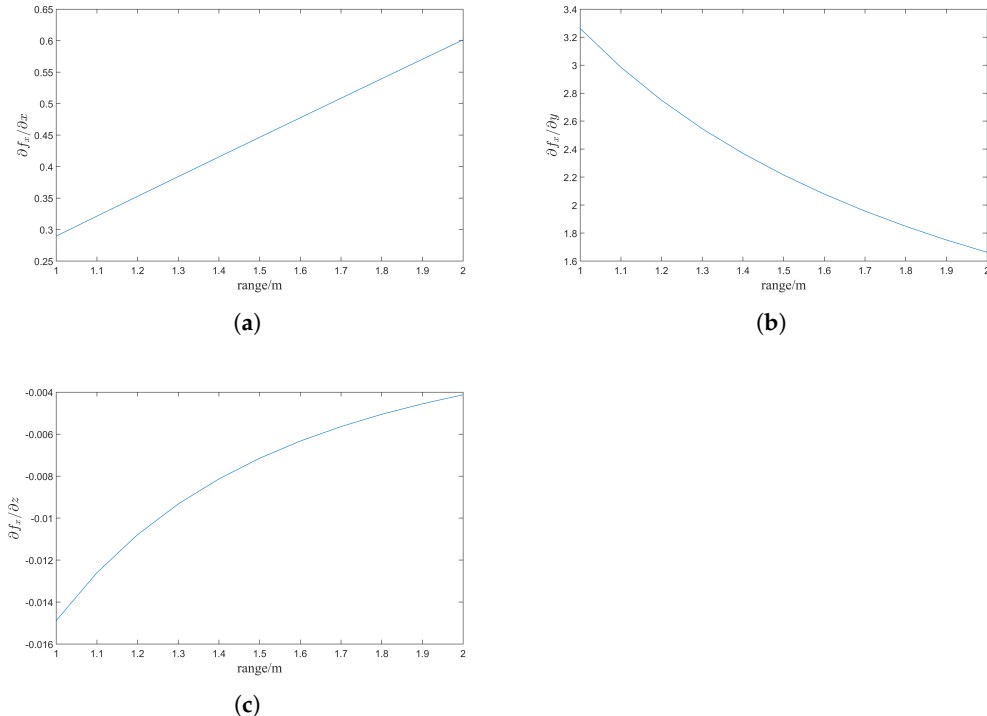

**Figure 8.** The first-order sensitivity coefficients with respect to (**a**) azimuth, (**b**) distance, and (**c**) altitude.

Set $\theta_z = 0.1° = 0.0017$ rad, $dx = 0.005$ m, $dy = 0.005$ m and $dz = 0.01$ m, so the deviation of the rotation angle $\theta_z$ is depicted in Figure 9. Compared to $\theta_z$, the estimation errors are less than 1% of the true value. Moreover, the errors do not exceed the required estimation accuracy in general. Results indicate that the EAFF algorithm exhibits a certain level of robustness.

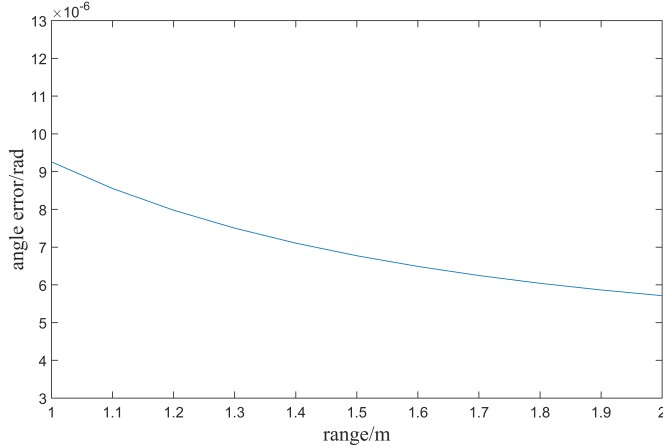

**Figure 9.** The trend of $\Delta\theta_z$ with the increasing distance.

## 5. The Experimental Results for Repeat-Pass THz InSAR

### 5.1. The Introduction of THz-SAR System

For the 300 GHz stepped-frequency (SF) SAR system, the block diagram of the measurement hardware system is shown in Figure 10, which depicts the two multiplier chains

and IF backend. The signal source generates two signals, namely the radio frequency (RF) signal and the local oscillator (LO) signal. These signals are divided into two channels using a power divider individually. The 50 MHz signal, down-mixed at mixer 1 by the RF and IF signals, is then multiplied by a factor of 18 to obtain a 900 MHz intermediate frequency (IF) signal. Then, the RF signal and the LO signal are utilized to transmit and receive signal separately after being multiplied by a factor of 18. The echo is down-mixed at mixer 2 to create the IF signal carrying the target information, and coherently demodulated with the 900 MHz reference IF signal to obtain the baseband echo, which is finally sampled and sent to the computer for processing. There have been some studies in THz-SAR imaging algorithms based on this radar, and the resolutions have achieved the millimeter level due to the large bandwidth [5,10].

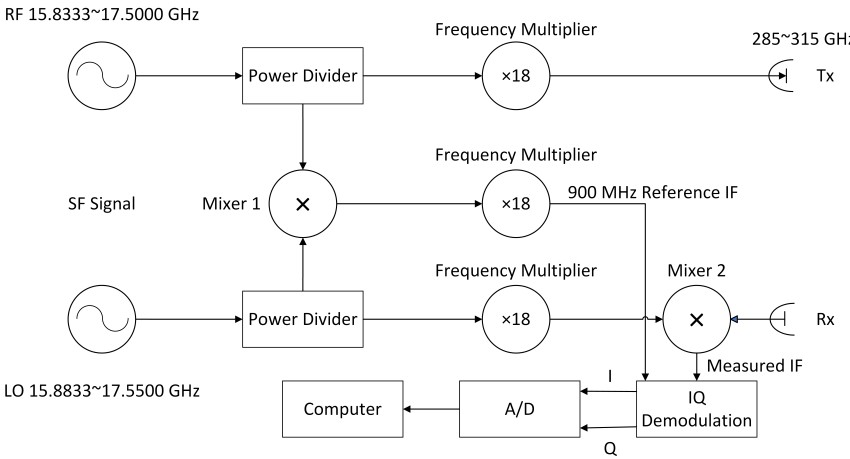

**Figure 10.** The block diagram of the 300G SF radar.

Figure 11 shows the imaging scene of the 300 GHz stepped-frequency SAR system, mounted on the track. During an operation, the vehicle's wheel meticulously controls pulses emission and reception for the precise and uniform sampling. This approach alleviates the issue of the non-uniform sampling during the signal acquisition process, leading to the enhanced quality of SAR imagery. The main parameters of the THz-SAR system are presented in Table 2.

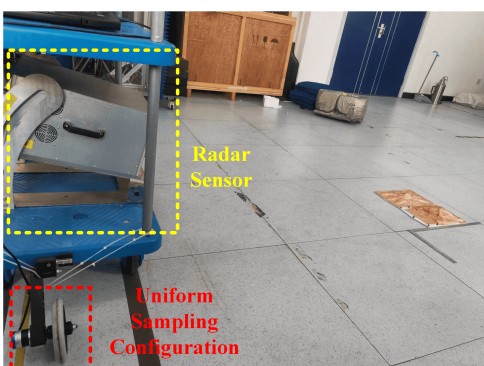

**Figure 11.** Optical photo of the THz-SAR system and imaging scenery.

**Table 2.** 300 GHz radar experimental parameters.

| Parameters | Value | Unit |
|---|---|---|
| Center frequency | 300 | GHz |
| Bandwidth | 28.8 | GHz |
| range solution | 5.2 | mm |
| Number of frequency steps | 1601 | |
| Frequency step | 18 | MHz |
| Azimuth beam width | 6.5 | (°) |
| Azimuth sampling rate | 412.8 | Hz |
| Transmitting power | 0.5 | mW |

### 5.2. The Experiment of the Knife Model

Initially, we should provide a detailed description of the experimental procedure. As Figure 12 shows, a mobile target approach was used to construct the baseline. Implementing repeat-pass interferometry with the radar mounted on a cart necessitates changing the relative distance between the targets and the radar. However, moving the radar in the cross-track direction to form a baseline poses significant challenges and undermines the precise control of a path. In practice, for laboratory targets, the relative distance can be altered by moving the scene. Therefore, using a fixed 1 mm precision ruler on the ground to calibrate the relative positions, the baseline could be constructed by positioning the scene at different locations. The received radar echoes are analogous to those collected from two distinct tracks, and the baseline error remains within an acceptable range.

After describing the way of acquiring the data of repeat-pass THz InSAR, the elevation of the experimental scene has been depicted in the following paragraph. Figure 12 illustrates the imaging scenario of the model knife. The calibration plane was fixed together with the model to facilitate mobility and maintain their relative positions. There is a noticeable height variation on the knife surface from the tip to the handle. At the back of the knife, the absolute height ranges from 1.4 cm to 1.8 cm, with a variation of 4 mm. At the blade, the absolute height ranges from 0.1 cm to 1.5 cm, with a variation of 1.4 cm. There is a sharp height transition at the handle, with a height difference of 5 mm at the back and 1 cm at the blade. By the way, the actual experiment was carried in a tilted plane to eliminate the strong reflection toward the radar, but the tilted angle was relatively small and did not make an impact on elevation estimation.

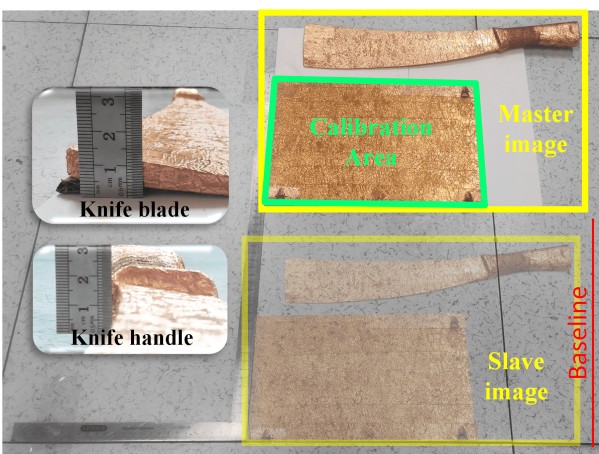

**Figure 12.** Optical photo of the imaging scenery of knife model. The baseline is constructed by scene transition.

We set $B = 0.3$ m and $\alpha = 0°$ for the baseline configuration. The interferogram of the knife model is depicted in Figure 13. It exhibits three prominent reflections, attributed to the corner reflectors characterized by significantly high scattering coefficients. The dB map

values for the blade portion are notably small, with the whole amplitude scale set from −180 to 0 dB, as shown in Figure 13a. Figure 13b presents the corresponding differential phase, displaying the distinct interference phase fringes on the calibration board, albeit with some minor noise. These fringes represent the rate of phase changes and manifest as an oblique pattern in the range and azimuth directions, signifying phase variations in both directions. The presence of fringes in the range direction is attributable to the interference effects of the electromagnetic waves and variations in the terrain height on the inclined surface, introducing the non-zero terrain phase. The fringes should not appear in the azimuth direction in the absence of variations. However, as analyzed in Section 3, EDF emerges in the azimuth direction as the radar follows a trajectory with a bias $\theta_z$.

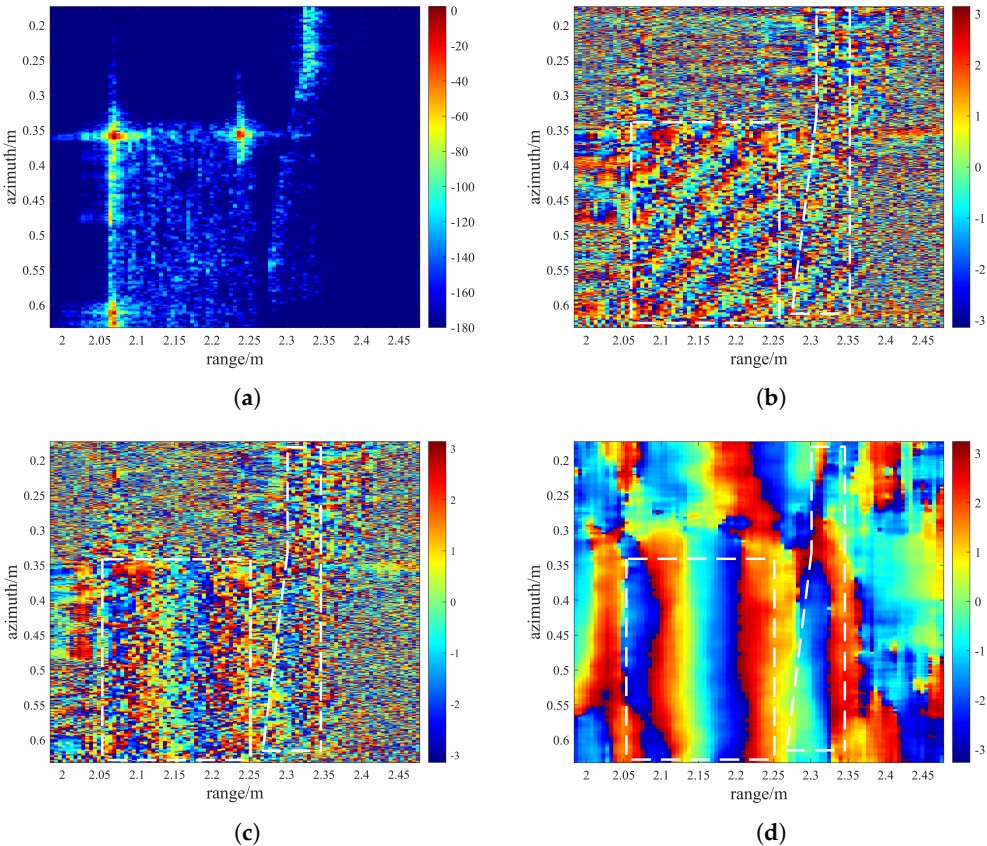

**Figure 13.** The interferogram of the knife model. (**a**) The interferogram magnitude. (**b**) The raw interferometric phase. (**c**) The interferometric phase calibrated by EAFF algorithm. (**d**) The filtered phase processed by circular filtering. The range encompassed by the white dashed line in the image corresponds to the location of the target.

In order to generate the DEM, the terrain phase needs to be obtained from the differential phase map. Figure 14 shows the frequency spectrum of the error phase fringe. The zero-padding is used and the oversampling number is increased to 4096 in order to improve the angle accuracy estimation. The azimuth frequency calculated by the FFT is −0.056 Hz, so one can obtain $\theta_z = -0.3836°$, according to (44) in this experiment. Figure 13c shows the terrain phase after removing the flat Earth phase and the calibration. Even if the angle deviation is smaller than −0.4°, there are the significant phase and elevation errors in the azimuth direction. This highlights the necessity of the baseline calibration algorithms. After the calibration using our EAFF algorithm, the phase fringes on the plane are clear, continuously decreasing, and exhibit the phase wrapping which corresponds to a gradual increase in height. Due to the severe noise interference, it is difficult for the human eye to discern the phase distribution in the knife section.

To obtain smooth 3D imaging, it is necessary to implement the phase filter. The classic circular mean filtering [29] is used, which has strong filtering capabilities and has been widely utilized. So we always employ the circular mean filter in our study for its robustness. $\phi$ and $\hat{\phi}$ represent the raw and filtered data, respectively. The output of the filtering is

$$\hat{\phi}(p,q) = \text{mean}_{m,n}\{\arg[\exp[j\frac{\phi(m,n)}{d(m,n)}]]\} + \arg[d(m,n)] \tag{57}$$

$$d(m,n) = \sum_{m=p-M}^{p+M}\sum_{n=q-N}^{q+N}\exp[i\phi(m,n)] \tag{58}$$

where $(p,q)$ is the position of a center pixel, and $(m,n)$ expresses the pixel position of a circular mean filter window. $d(m,n)$ represents the average of $\phi(m,n)$ in an $M*N$ window centered at $(p,q)$. The window sizes are chosen according to multiples of the resolutions greater than 1. Considering the azimuth oversampling of the radar, the azimuthal window size is 15 pixels and the range is 3 pixels. After filtering, the low frequency part of the interferometric phase is well extracted. As Figure 13d shows, the filtering significantly reduces residual points, and the filtered phase is more suitable for the subsequent phase unwrapping and height extraction processes.

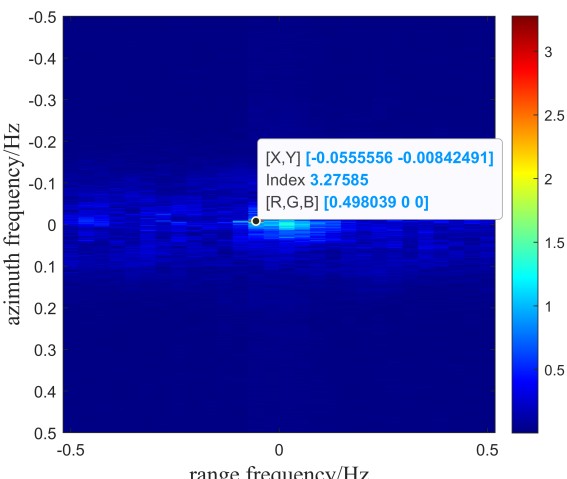

**Figure 14.** The frequency spectrum of phase fringe on calibration plane.

Finally, the 3D images processed in three ways are shown in Figure 15. As Figure 15a shows, 3D reconstruction of the knife model can be barely identified without calibration. The details of the height distribution on the target become even more difficult to discern. Figure 15b illustrates the imaging outcomes following the calibration using the GCPs. The calibration process involved ten iterations, and the estimation result is $B = 0.2987$ m. It is evident that the millimeter-level error falls short of the generation of the high-precision DEM. On the contrary, as shown in Figure 15c, from a overall perspective, there is a clear upward trend in height on the blade surface, with smooth height variations, allowing for a crisp depiction of the 3D shape of the entire model. Along the azimuth direction, the height change at the blade tip is 1.8 cm, while at the handle protrusion is 1.5 cm. Similarly, at the back of the knife, the measured height change is 8 mm, and at the handle protrusion it is 1.3 cm, consistent with the millimeter-scale precision. Due to the inclination of the scene, the radar is more capable of detecting, bringing about the high-quality data at the handle position and thus enhancing the 3D contour of the object. As THz InSAR provides a elevation accuracy at the millimeter level, a highly distinct three-dimensional structure is observed at the handle position, demonstrating the ability of THz InSAR to achieve higher precision in three-dimensional imaging.

### 5.3. The Experiment of the Terrain Model

Previously, 3D reconstruction was carried out using the relatively simple knife model. The results indicate that the proposed EAFF algorithm significantly improves the DEM accuracy. Considering the lower richness of three-dimensional information of the knife model, the terrain model is built to validate the high-precision elevation accuracy of InSAR in the THz band. Also, the data acquisition process remains consistent with the procedure mentioned earlier.

A terrain undulating scenario is constructed, as shown in Figure 16. This scenario consists of two main parts: the terrain area and the calibration area. The calibration area is composed of the flat terrain and corner reflectors, which are utilized for the different calibration methods. The baseline calibration algorithm based on the GCPs is employed for the comparison with our EAFF algorithm. The terrain area comprises Area 1 and Area 2, both exhibiting undulating topography. Area 1 has a maximum height of 3 cm and covers an area of $0.3 \times 0.2$ m$^2$, while Area 2 reaches a maximum height of 4 cm and spans an area of $0.2 \times 0.2$ m$^2$, with steeper undulations compared to Area 1.

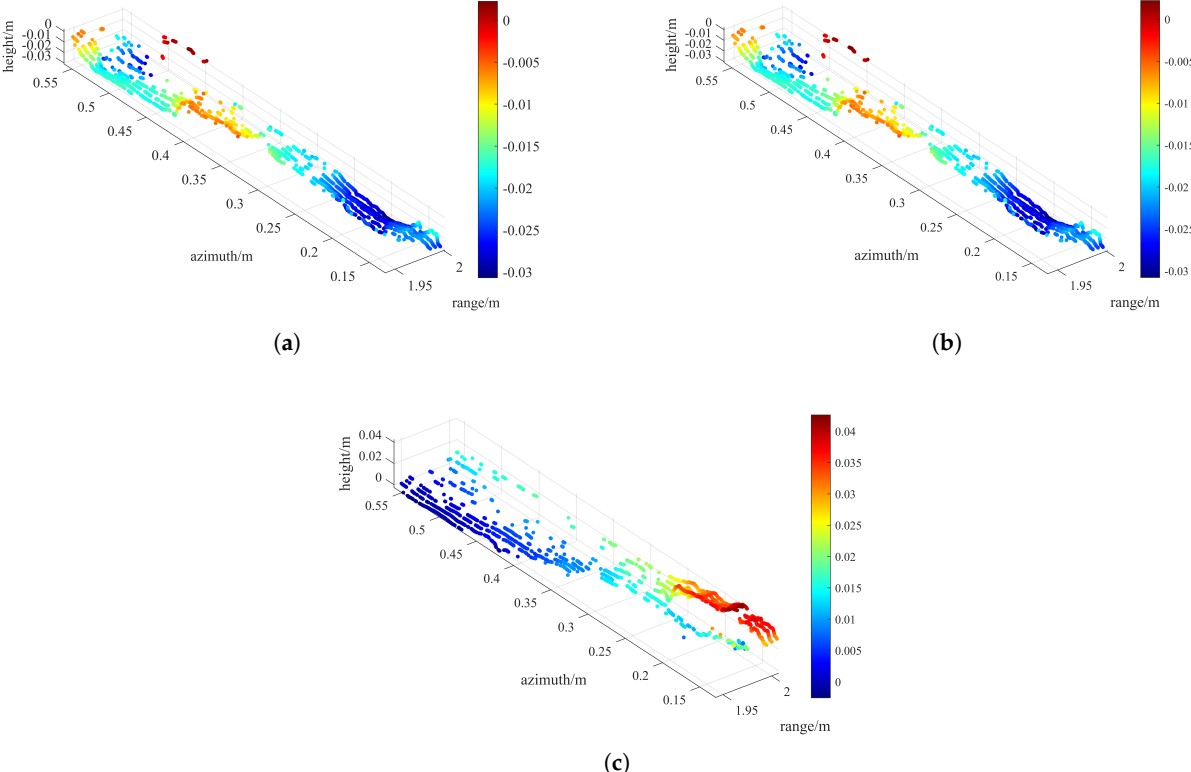

**Figure 15.** Three-dimensional reconstruction of the knife model (**a**) without calibrating, (**b**) calibrated based on GCPs, and (**c**) with calibrating by the EAFF algorithm. High-intensity points in the interferogram are selected as valid data for three-dimensional visualization. Different colors represent different heights, allowing for clear visualization of the object's structure and height variations.

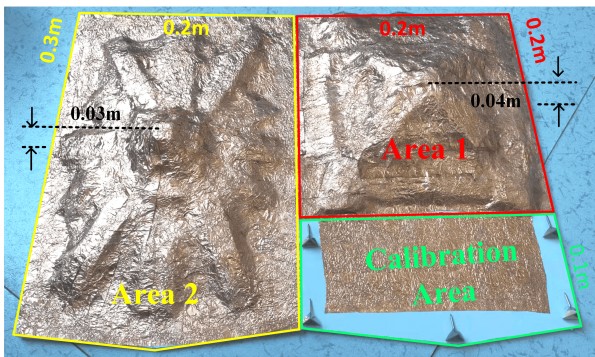

**Figure 16.** Optical photo of the imaging scenery of terrain model.

Taking into account the requirements for the height sensitivity and registration, we set $B = 0.09$ m and $\alpha = 0°$. Figure 17 illustrates the interferometric processing steps. In Figure 17a, the interferometric amplitude image shows that the azimuthal range extends from 0.2 to 0.58 m, and the range in distance spans from 1.35 to 1.6 m, with a dynamic dB range from $-10$ to 0 dB. The regions below $-10$ dB are regarded as no target, likely due to the energy obstruction by the foreground terrain. The five positions with stronger reflections are also the corner reflectors, which are used for the baseline calibration the GCPs.

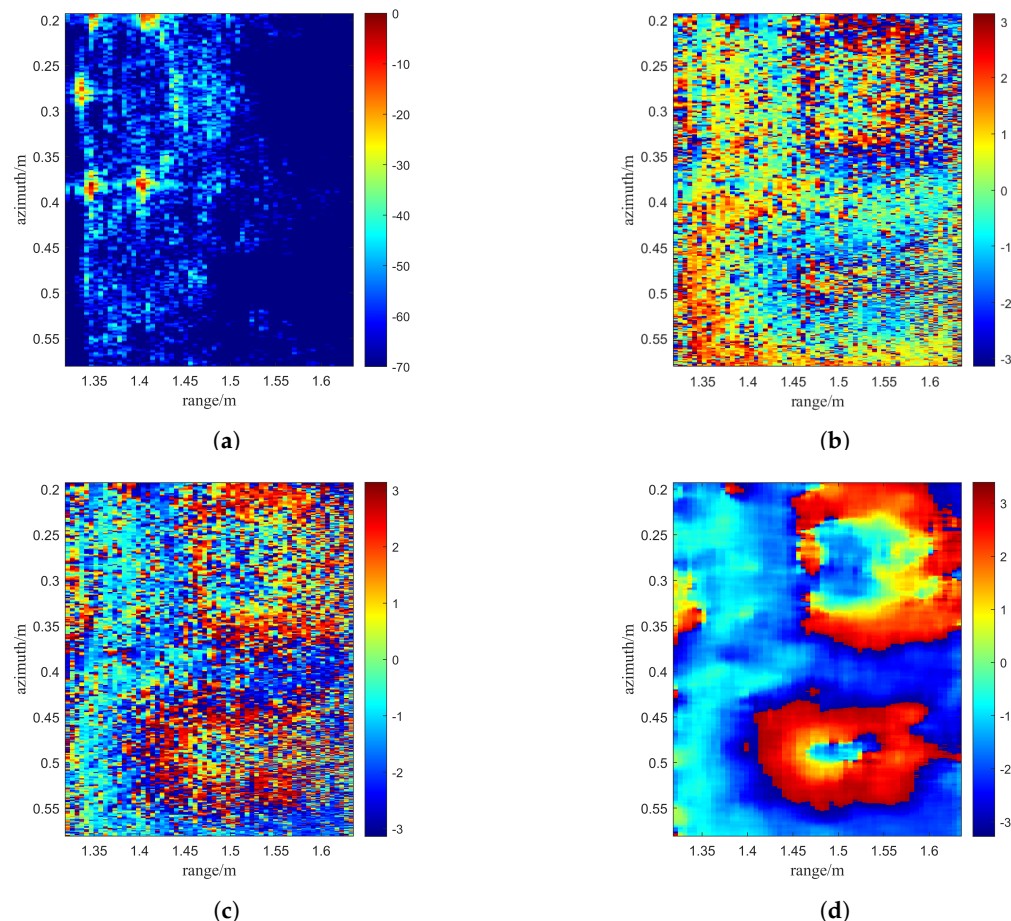

**Figure 17.** The interferogram of the terrain model. (**a**) The interferogram magnitude. (**b**) The raw interferometric phase. (**c**) The interferometric phase calibrated by EAFF algorithm. (**d**) The filtered phase processed by circular filtering.

Figure 18 shows the frequency spectrum of the error phase fringe in the terrain model experiment. The processing is consistent with the knife experiment. The azimuth frequency

calculated by the FFT is $-0.045$ Hz, so one can obtain $\theta_z = -0.02°$, according to (44). The rather small angle results in millimeter elevation errors throughout the entire image. Figure 17b,c represent the interferometric phase images before and after the calibration in order. It can be observed that, after the compensation, the phase errors in different azimuth directions are reduced. Figure 17d shows the filtering phase, and the stronger noise has been eliminated. However, the comparison between the two terrain areas reveals that the circular filtering is less effective in situations with dense fringes. Preserving the continuity of phase edges is a crucial study for the phase filtering of THz InSAR in the future.

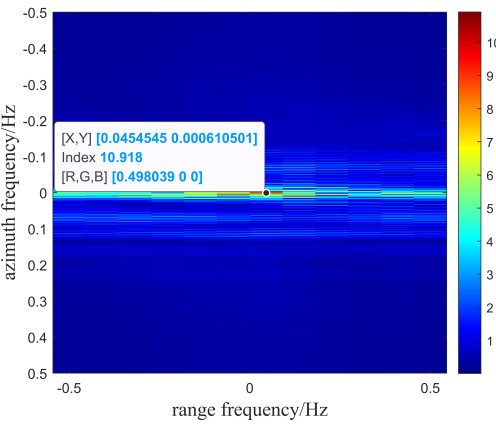

**Figure 18.** The frequency spectrum of phase fringe of the terrain model experiment.

Figure 19 illustrates the outcomes produced by three distinct operations. The four specific points with the same 2D ordinates are labeled for the detailed analysis of the three processes. As Figure 19a shows, without calibration, a discernible three-dimensional contour is still evident. The reason is that the angle error is relatively small compared to the knife experiment. The maximum absolute elevation of Area 1 is 0.0298 m, and that of Area 2 is 0.0462 m. The elevation disparity between the two mountain-like features is calculated by $0.0462 - 0.0298 = 0.0164$ m, which includes a 0.0064 m deviation from the true value. Following the baseline calibration with the GCPs, Figure 19b shows that the elevation deviates even further from the actual values. The height difference between the highest points of two areas shrinks to 0.0027 m. The elevation discrepancy is accentuated by varying errors in the radar sampling path, and the baseline accuracy calibrated based on the GCPs is insufficient. Figure 19c showcases the result obtained through the proposed EAFF algorithm. The elevation variation between the two features has been reduced to $0.0558 - 0.0447 = 0.0111$ m, so the elevation bias has been decreased to 0.0011 m by calibrating with the EAFF algorithm. The experiment indicates its effectiveness in reducing relative errors within a certain scene.

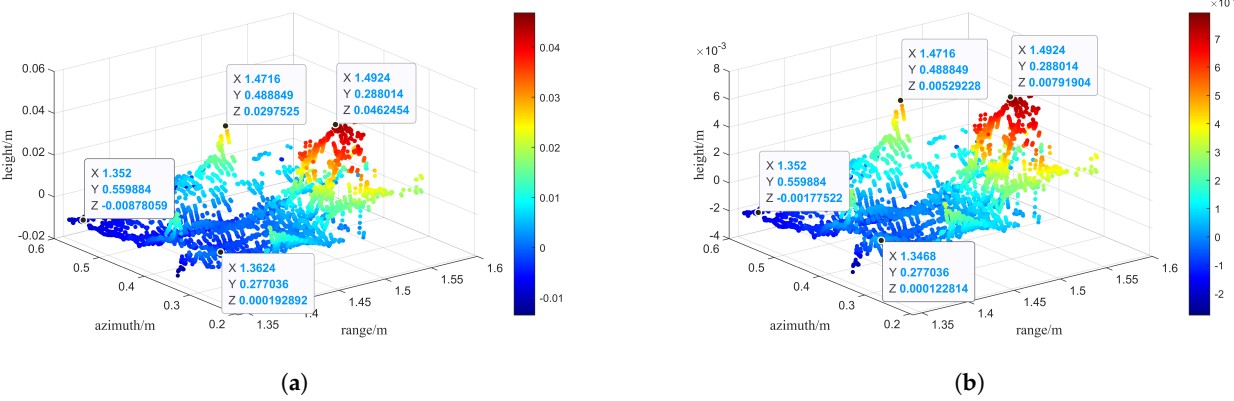

(**a**)                                                                 (**b**)

**Figure 19.** *Cont.*

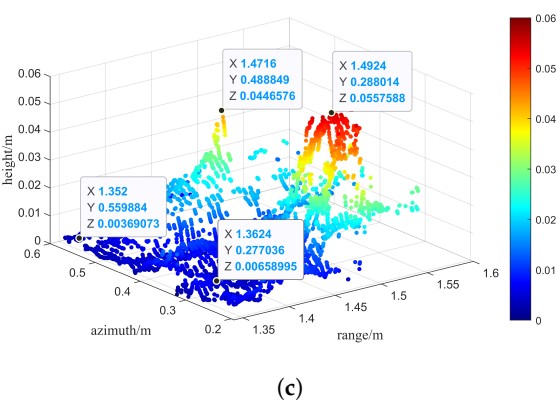

(**c**)

**Figure 19.** Three-dimensional reconstruction of the knife model, (**a**) without calibrating, (**b**) calibrated based on GCPs, and (**c**) with calibrating by the EAFF algorithm. Different colors represent different heights, allowing for clear visualization of the object's structure and height variations.

## 6. Conclusions

In our study, we introduce InSAR imaging technology to the terahertz band, and propose the EAFF method for the high-precision baseline calibration for THz InSAR. Firstly, based on the scale-down imaging geometry, we conducted a comprehensive theoretical analysis of principles, coherence, and elevation accuracy of high-precision THz InSAR imaging. Secondly, according to the required elevation precision for THz InSAR, we proposed the EAFF method for the high accuracy baseline calibration to obtain the high-precision DEM in THz InSAR. Initially, we formulated the model to account for the non-parallel sampling path errors in repeat-pass THz InSAR, and conducted the theoretical analysis of the phase errors induced by the non-parallel errors. To further this analysis, we leveraged a reference DEM to establish a connection between the azimuth fringe frequency of EDF and the repeat-path angle bias. By transforming the position error into the frequency spectrum estimation, the influence of unknown SAR sampling positions was mitigated. In accordance with the desirable elevation accuracy, the estimation accuracy of frequency spectrum, a rotation angle, and a baseline could be calculated exactly. The simulations at varying noise levels showed that the EAFF method based on the FFT can achieve the theoretical accuracy when $\gamma > 0.3$, and this is easily satisfied in most baseline configurations. In the end, we configured repeat-pass THz InSAR system with the 300 GHz stepped-frequency radar. The application of our EAFF calibration method significantly enhanced the DEM accuracy for both the knife and terrain models. These results demonstrate that the EAFF baseline calibration method is well suited for THz InSAR and significantly contributes to obtaining high-precision interferograms, thereby advancing further research in the field of high-precision THz InSAR imaging.

By means of the high accuracy of THz SAR, the exploration of InSAR in the THz band is of great value to the short-range three-dimensional imaging. For example, the high-precision detailed map of surroundings will improve the ability of smart cars for the higher spatial sensing and navigation. However, in the context of phase filtering and unwrapping algorithms, the substantiation in both theory and experiments of THz InSAR is imperative to enhance the precision of the phase retrieval in THz InSAR interferograms. Moreover, the condition of our study is that a signal phase and a target range exhibit a linear relationship. The mathematical model for remote sensing in the terahertz band needs to be deeply studied in the future.

**Author Contributions:** Conceptualization, Z.W.; Methodology, Z.W.; Software, Z.W. and G.Z.; Validation, Z.W.; Data curation, G.Z. and S.Z.; Writing—original draft, Z.W.; Writing—review & editing, C.L.; Supervision, C.L., S.Z., X.L. and G.F.; Funding acquisition, C.L., X.L. and G.F. All authors have read and agreed to the published version of the manuscript.

**Funding:** This research was funded by the National Natural Science Foundation of China under grant 61988102, grant 61731020, and the National Key Research and Development Program of China under grant 2018YFF01013004, and the Beijing Municipal Natural Science Foundation under grant L223007, and the Key-Area Research and Development Program of Guangdong Province under grant 2020B0101110001, and the Project of Equipment Pre-Research under grant WJ2019G00019.

**Data Availability Statement:** No new data were created or analyzed in this study. Data sharing is not applicable to this article.

**Conflicts of Interest:** The authors declare no conflict of interest.

**Appendix A**

In order to convert (3) into standard form of the cubic equation, make

$$x = \delta\theta, a = 1, b = \tan(\theta - \alpha), c = -6, d = -\frac{6\phi_h}{kB\cos(\theta - \alpha)}$$

To solve a cubic equation, it is necessary to introduce a variable and transform the cubic equation into a quadratic equation, then let

$$x = z - \frac{b}{3}, p = c - \frac{b^2}{3}, q = \frac{2b^3}{27} - \frac{bc}{3} + d$$

where

$$p = 36 - 3\tan(\theta - \alpha)$$

$$q = 2\tan^2(\theta - \alpha) + 6\tan(\theta - \alpha) + \frac{6\varphi_{unwrap}}{kB\cos(\theta - \alpha)}$$

and get the equation $z^3 + pz + q = 0$. Then, substitute $z = u + v$ to cubic equation of $z$ to gain

$$u^3 + v^3 = -q, u^3v^3 = -\frac{p^3}{27}$$

then the quadratic equation in terms of $X$ is given by

$$X^2 + qX - \frac{p^3}{27} = 0, \quad p, q \in \mathbb{R}$$

where $U = u^3$ and $V = v^3$ are two complex roots.

Through computation, it is often found that the discriminant

$$\Delta = q^2 + \frac{4p^3}{27}$$

is less than 0, indicating that the equation has three complex roots, which share the same expression

$$z_k = 2\sqrt{-\frac{p}{3}}\cos\left(\frac{\theta + 2k\pi}{3}\right) \in \mathbb{R}$$

$$\theta_k = \frac{1}{3}\arccos\left(\frac{3q}{2p}\sqrt{-\frac{3}{p}}\right) - \frac{2k\pi}{3}, k = 0, 1, 2.$$

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
