# Peer review of "A High-Precision Baseline Calibration Method Based on Estimation of Azimuth Fringe Frequency with THz Interferometry SAR"

_remotesensing, doi:10.3390/rs15245755_

Round 1
Reviewer 1 Report
Comments and Suggestions for Authors
Manuscript entitled “A High Precision Baseline Calibration Method Based on Estimation of Azimuth Fringe Frequency (EAFF) with THz Interferomery SAR" shows that the authors introduced an innovative baseline calibration method rooted in fringe frequency estimation, and the attainment of a high-precision baseline accuracy is attributed to the amalgamation of InSAR techniques with terahertz waves. The theory can also be substantiated through simulations and experiments. There are some detailed problems in this paper. It may be suitable for publication in Remote Sensing after some detailed revisions. The comments are as follows:
1. The literature review is currently insufficient. Please include a comprehensive discussion of existing baseline calibration methods in the introduction section.
2. A thorough discussion of the method's applicability conditions is not concluded in this paper. Please furnish additional details concerning the prerequisites for its deployment.
3. In line 121 and 142, the referencing of the figures is not accurately presented. Please rectify them.
4. In line 171, the “s1 and s2” are not consistent with the expressions in Equation (16) and (17). Please amend them.
5. In Figure 5, the horizontal axis is not labeled. Please provide it.
Comments on the Quality of English Languagethe english is fine to me.
Author Response
Dear Reviewer:
We gratefully appreciate for your valuable suggestions and careful check. We have read through comments carefully and have made corrections. The attached PDF document has been provided for our revisions. Looking forward to hearing from you soon.
Sincerely,
Zeyu Wang

Reviewer 2 Report
Comments and Suggestions for Authors
A High Precision Baseline Calibration Method Based on Estimation of Azimuth Fringe Frequency (EAFF) with THz Interferomery SAR
Authors are attempting to demonstrate a more effective estimate of elevation of land features (DEM) by using the terahertz band of synthetic aperture radar data. Instead of ground control points the elevation data is calibrated using azimuth fringe frequency (EAFF) of interferometric phase.
Research articles which might be included on THz SAR
https://ieeexplore.ieee.org/abstract/document/10111030
https://ieeexplore.ieee.org/abstract/document/9567297
http://journal.sitp.ac.cn/hwyhmb/hwyhmben/article/abstract/2021203
Line 81 – “heihgt error”
Line 112 – “Generally speaking, Ï• is consist of the terrain phase” example of grammar errors that should be fixed by native speaker
Line 121 and Line 142 –“shown in Figure ??,” figure number not found (probably figure 1).
Line 292 – use of EAFF is a rather ingenious way to overcome angle error problems. Math seems correct.
Line 305 – “plane ,represented” comma error
Line 350 – “0.3 THz stepped-frequency (SF) SAR” isn’t this just a 300 GHz system? Research into Gigaherz suggests 1 to 35 is normal range for this application so this seems new. Is it convention to talk about Terraherz when it’s still under 1?
Line 477 – Experiments are effective in showing higher resolution. However, why is the more complex model terrain shaped in practice? It implies in my mind that this would be used from elevation (drone mapping) rather than short distance (smart car / robot sight) applications. I then question if small scale models like this vs the distance from source / size of the waves to object mapped truly reflect a drone at 3000 ft trying to map terrain.
Comments on the Quality of English LanguageThe English wording of many sentences is awkward, and I would suggest a native language speaker edit the structure.
Author Response
Dear Reviewer:
We appreciate for your rigorous comments, and we have read through comments carefully and have made corrections. The attached PDF document has been provided for our revisions. Looking forward to hearing from you soon.
Sincerely,
Zeyu Wang

Reviewer 3 Report
Comments and Suggestions for Authors
see attached file

Author Response
Dear Reviewer:
We have carefully considered all comments you mentioned and revised our manuscript accordingly. The attached PDF document has been provided for our revisions. Looking forward to hearing from you soon.
Sincerely,
Zeyu Wang

Round 2
Reviewer 3 Report
Comments and Suggestions for Authors
Please correct line 214 asterisk and line 218 b and add the correspondent reference , still missing
Author Response
We are sorry for our mistakes, and the references have been cited correctly in the revision.

![]()